# Faulting patterns in the Lower Yarmouk Gorge - potentially influence groundwater flow paths

Inbar Nimrod[1,2,3], Eliahu Rosenthal[1], Fabien Magri[4,5], Marwan Alraggad[6], Peter Möller[7], Akiva Flexer[1], Joseph Guttman[8], Christian Siebert[9]

[1] Porter School of the Environment and Earth Sciences, Tel Aviv University, Israel
[2] Department of Geophysics and Space Sciences, Eastern R&D, Ariel, Israel
[3] Department of Physics, Ariel University, Ariel, Israel
[4] UFZ Helmholtz Centre for Environmental Research, Dept. Environmental Informatics, Leipzig, Germany
[5] Freie Universität Berlin, Chair of Hydrogeology, Berlin, Germany
[6] Water, Energy and Environment Center, University Of Jordan.
[7] Helmholtz Centre Potsdam German Research Centre for Geosciences - GFZ, Section 3.4, Telegrafenberg, Potsdam, Germany
[8] Mekorot, the National Water Company, Hydrology Department, Tel Aviv, Israel
[9] UFZ Helmholtz Centre for Environmental Research, Dept. Catchment Hydrology, Halle, Germany

*Correspondence to*: Nimrod Inbar (inbar.nimrod@gmail.com)

**Abstract.** Recent studies investigating groundwater parameters e.g., heads, chemical composition, and heat transfer, argued that groundwater flow paths in the Lower Yarmouk Gorge (LYG) area are controlled by geological features such as faults or dikes. However, the nature of such features, as well as their exact locations, were so far unknown. In the present manuscript, we propose a new fault pattern in the LYG area by compiling and revising geological and geophysical data from the study area including borehole information, geological maps cross-sections and seismic data from southern Golan Heights and northern Ajloun Mountain. The presented pattern is composed of strike-slip and thrust faults, which are associated with the Dead Sea Transform system and with the Kinnarot pull-apart basin. Compressional and tensional structures developed in different places forming a series of fault-blocks probably causing a non-uniform spatial hydraulic connection between them. This study provides a coarse fault block model and improved structural constraints that serve as fundamental input for future hydrogeological modeling which is a suggested solution for an enigmatic hydrological situation concerning three riparian states (Syria, Jordan, and Israel) in a water scarce region. In areas of water scarcity and transboundary water resources, transient 3D flow simulations of the resource are the most appropriate solution to understand reservoir behavior. This is an important tool for the development of management strategies. However, those models must be based on realistic geometry, including structural features. The study at the LYG is intended to show the importance of such kind of structural investigations for providing the necessary database in geologically stressed areas without sufficient data. Furthermore, during the hydrogeological investigation, a mismatch with results of pull-apart basin rim faults evolution studies have been discovered. We argue that this mismatch may result from the settings at the eastern rim of the basin as the en-echelon changes from pull-apart basins (Dead Sea, Kinnarot, Hula) to a push-up ridge (Hermon).

**Introduction**

The Lower Yarmouk Gorge (LYG) is a prominent geomorphological feature located in the triangle between Israel, Syria and Jordan, east of the Dead Sea Transform (DST) between the Sheikh Ali fault to the north (in the central Golan Heights, henceforth called GH) and the Zarka fault to the south, in northwest Jordan (Fig. 1). Structurally, the LYG is located along the southern extremity of the Golan syncline bounded by Mt. Hermon in the north and by the Ajloun dome in the south (Mor, 1986). The

hinge line of the Golan syncline is considered to be located along the Sheikh Ali tensional fault zone, a few km north of the LYG (Meiler, 2011; Shulman et al., 2004).

    The LYG drains the natural flow of the 6,833 km$^2$ large surface catchment of the Yarmouk River basin (Fig. 1) and flows into the Jordan River south of the Sea of Galilee. Additionally, the LYG is receiving groundwater flows from the Ajloun Dome (El-Naser, 1991), from Mt. Hermon through deep aquiferous

formations in the subsurface of the Golan Heights (Gvirtzman et al., 1997) and from the Hauran Plateau (Siebert et al., 2014), situated to the NE.

    Through the springs of Hammat-Gader and Mukheibeh and the wells of Mukheibeh (henceforth abbreviated as MU) and Meizar (henceforth abbreviated as MI), all located along the LYG, groundwater emerges with different hydraulic heads, chemical compositions, and temperatures (Siebert et al., 2014).

Most studies of the springs emerging on both sides of the gorge were carried out between 1970 and 1990's and considered mainly their geochemical and hydrological characteristics and concluded that their outflow is a mixture of shallow, cold and fresh groundwater with a deep water body of higher salinity and temperature (Arad and Bein, 1986; Arad et al., 1986; Baijjali et al., 1997; Eckstein and Simmonsi, 1977; El-Naser, 1991; Mazor et al., 1973; Mazor et al., 1980). However, the mechanisms and pathways,

allowing the deep hot brine to ascend remained obscure. Deep-reaching faults below the area of the LYG would be promising structural features, controlling such hydrogeological systems.

    Recent studies attempted to explain the mechanisms responsible for the outflow of hot springs and high-pressure groundwater identified in wells located along the LYG. Roded et al. (2013) applied a conceptual structural model of the Golan syncline to numerically simulate regional groundwater flow and heat

transfer, introducing a 5 km wide zone of enhanced vertical permeability below the LYG. Based on geochemical evidence, Siebert et al. (2014) suggested that the groundwaters in the LYG are fed by water originating from (a) Mt. Hermon, (b) Northern Jordan and (c) the Hauran Plateau in Syria, and do not mix arbitrarily, which suggests the existence of a zone of hydraulic anisotropy along the LYG. At that weakness zone, brines, likely heated by buried dikes may ascend and feed the Hammat Gader springs

and some boreholes in the central LYG.

    Numerical modeling of a 2D section crossing the LYG inferred the existence of a fault in the gorge and revealed a complex groundwater flow pattern in the study area (Magri et al., 2015). This was followed by a 3D model of a hypothetical fault tracing along the LYG and suggested a mechanism of heat-driven convection cells (Magri et al., 2016). Consequently, the anomalous heat flow was studied by a method

of inverse problem, suggesting that the thermal constraints of the system require one of two scenarios, (a) relatively permeable and continuous faults, cutting through the entire geological section, reaching at depth the Triassic beds, or (b) local fractures interconnected by a highly permeable Cretaceous aquifer (Goretzki et al., 2016). Tzoufka et al. (2018) could substantiate the existence of a zone of hydraulic

anisotropy along the LYG, possessing high hydraulic conductivity along its course, while S⇔N oriented groundwater migration across the plane is impeded. However, the above-mentioned studies are based on simplified assumptions of fault locations and orientation.

**Stratigraphy**

The exposed stratigraphy of the Golan Heights reveals mostly Pliocene to Quaternary basalts (Mor, 1986; Heimann et al., 1996; Dafny et al., 2003). In the north, the entire sequence from Quaternary basalts to Jurassic limestone is exposed close to Mt. Hermon (Hirsch, 1996; Picard and Hirsch, 1987). In the central part of the Golan Heights and along its southern margins Eocene to Miocene sediments are exposed in the wadis flowing towards the DST (Michelson, 1979). Middle Cretaceous rocks crop out in the Ajloun Dome, revealing a Campanian formation which does not exist either west of the DST (Flexer, 1964) nor in the northern part of the GH. This formation is known by the Jordanian nomenclature as the Amman Silicified Limestone (ASL) or - by its hydrological term - the B2 Aquifer (Andrews, 1992; El-Naser, 1991).

By its distinctive lithology of silicified limestone and chert, the Campanian ASL formation is easily identified in outcrops and boreholes. It overlies the Santonian (B1) limestone and is overlain by a thick Maastrichtian (B3) layer of marl. The transition to the underlying Santonian (B1) is somehow difficult to identify because the B1 unit is a thin layer (30-50 m) of limestone without cherts. However, the occurrence of dolomite or of dolomitic limestone clearly defines the Turonian (A7). Contrarily to the hydrological characteristics of the Senonian aquiclude in central and northern Israel, the ASL (B2) in northern Jordan makes up the upper part of the most essential aquifer in Jordan. The MI-1, MI-2 & MI-3 boreholes drilled in the southern Golan up to a distance of 6 km north of the LYG, reveal a lithostratigraphic sequence, similar to the Ajloun, confirming that the units continue across the gorge. South of the Ajloun Dome, in Wadi Zarka (Fig. 1), the exposed stratigraphy reveals the lithostratigraphic sequence down to Jurassic beds. The full section down to the Precambrian basement occurs in several deep boreholes drilled in northern Jordan and within the eastern escarpment of the Lower Jordan Valley (Abu-Saad and Andrews, 1993). In the southern GH, MI-1 is the northernmost borehole drilled to Turonian, beds providing a complete section of the overlaying lithology.

**Tectonics**

The Kinnarot basin is a link in the chain of pull-apart basins scattered along the DST formed by the left lateral movement along the Sinai – Arabian Plate boundary that started during Early to Middle Miocene (Garfunkel, 1981; Gvirtzman and Steinberg, 2012). The transform itself was located on the eastern side of the basin by seismic interpretation showing the compressional structure of the Tel-Qatzir elevated block (Inbar, 2012). To the north, Mt. Hermon manifests a shift in the en-echelon arrangement from left- to right-stepping resulting in a restraining geometry (Weinberger et al., 2009) which causes uplifting of the Lebanon and Anti-Lebanon mountains (Beydoun, 1977) as well as the deepening of the Golan syncline. The LYG, the southern GH and the Northern Ajloun, located along the eastern rim of the Kinnarot basin, are considered to be subject to the regional forces applied by this active plate boundary.

Models of pull-apart basins with different scales and from various places around the world (e.g. Corti and Dooley, 2015; Sagy and Hamiel, 2017; Smit et al., 2008; 2010; Sugan et al., 2014) address mostly the shape of their boundaries and internal processes. However, these models predict fault evolution at the rims as well. Such a scenario as the collision of Kinnarot basin's eastern rim with Mt. Hermon has not been modeled to the best of our knowledge. The conspicuous Shaikh Ali fault at the NE corner of the basin (Fig.1) is a clear indication for this different tectonic setting. Therefore, the Kinnarot basin seems to provide a unique opportunity to explore the effect of such a setting on fault evolution at pull-apart basin rims.

Along the eastern side of the DST, strike-slip faults are known to branch out and penetrate inland, directed NE-SW and E-W, across its rims (Fig. 1; Andrews 1992; Shulman 2004). Though the LYG is a prominent morphological feature east of the DST, its structure was never explored by continuous land geophysics (e.g. seismic, electric, etc.) due to its function as an international border. Consequently, attempting to bridge the gap in data resulted only in theoretical and qualitative models.

Considering the extreme thickness difference between Jurassic sections in Mt. Hermon and in the Ajloun (2,000 m and 400 m, respectively) and by comparing seismic lines from southern GH and from the Coastal Plain of Israel, Shulman et al. (2004) suggested the existence of a major fault tracing along the LYG. However, based on interpolation between the results of the interpretation of seismic data in the southern GH and deep borehole data in Jordan, it was suggested that the thickness of the Jurassic rock-sequence increases gradually from the south northwards and therefore a major fault in the gorge area should not be inferred (Meiler, 2011). The present study shows that although there is no evidence for large vertical displacements at the surface, seismic and borehole data indicate strike-slip faults crossing the LYG forming fault-blocks. Therefore, such faults should be taken into consideration when discussing groundwater hydrology.

**Data and methods**

The current study relies on reviewing, compiling and evaluating available geological and geophysical data from southern Golan and northern Ajloun. Most of the seismic data were collected at the Golan Heights during the 1980s (Shulman and May, 1989), published later as time migrated data analyses (Shulman et al., 2004) and reprocessed by Meiler (2011) using PSDM (Pre-Stack Depth Migration) to analyze regional aspects. In the current study, we aimed to refine Meiler's faulting interpretation of PSDM line DS-3545 (Figs. 1; 2) to shift the precision for a less broad perspective (i.e. Meiler neglected the westernmost fault in the regional context, which might be relevant at a local scale). Additionally, seismic data from southern Golan (Mevo Hamma; Fig. 1) (Shulman et al., 2004; Bruner and Dekel, 1989; Meiler, 2011) were reinterpreted to considerably improve the structural information in the vicinity of the Lower Yarmouk Gorge. Based on these data, in combination with lithological data from boreholes (Table 1), a new geological cross section was constructed along the LYG, running SW-NE (Fig. 3). Faults from surface mapping (Michelson, 1972), reinterpreted seismic lines and geological profiles (Sahawneh, 2011) were considered in order to generate an aerial view of faults and of fault-block patterns in the study area based on the tectonic concept of pull-apart basin rims. Such models predict evolution of different types

155 of faults at the margins of the main basin mainly according to its maturity, size and symmetry (obliqueness) (Wu et al., 2009; Rahe et al., 1998; Sugan et al., 2014; Smit et al., 2008; Smit et al., 2010). Finally, the previously mapped faults were traced together with those identified during the present study. It is important to note that the fault lines drawn on aerial view maps are a representation of an actual near-surface fault zone which converges to a single deep root as illustrated by seismic interpretation and

160 the geological cross-section.

## Results and Discussion

### *Reinterpretation of seismic data*

The ENE-WSW trending and 17 km long seismic line DS-3545 (Fig. 2a) runs parallel to the LYG, about 6 km north of it. Reinterpretation of the pre-stack depth migration (PSDM) conducted by Meiler (2011),

165 shows an additional flower-structure fault at its SW part indicating a set of strike-slip faults crossing that line. Although little to no horizontal displacement is visible on that seismic section, folds between the flower-structure fault branches clearly indicate lateral displacement. Moreover, the deep roots of the traced strike-slip faults suggest that it is related to significant regional tectonics. Its proximity to the DST combined with its effect on shallow lithology may advocate its connection to the DST tectonics.

170 Seismic line GP-3662 (Fig. 2b) trends NE-SW and is 4.5 km long. Its SW end is located near MI- 2. The time migrated line described by Bruner and Dekel (1989) was re-evaluated considering the results from the reinterpretation of seismic line DS-3545 and MI-1 and MI-2 stratigraphy. Near top Turonian marker was adjusted to fit its depth in the nearby Meizar wells and DS-3545 seismic line by depth to time conversion using RMS velocities of the original GP-3662 time migrated section. The near top Turonian

175 reflector (Fig. 2b, green markers) is interrupted by a thrust fault indicating compressional stress along the line. The different dip angles of shallower reflectors (Fig. 2b, yellow markers) at the SW side of the line as well as folded reflectors at the upper part of the section support that idea.

Surface and shallow geological data in the southern GH and northern Ajloun indicate possible surface and shallow fault patterns (Sahawneh, 2011; El-Naser, 1991; Michelson, 1979). All this data suggests

180 the existence in the study area of short faults of limited vertical displacement. However, deep seismic data were collected in the southern GH (Shulman et al., 2004; Meiler, 2011) revealed that normal faults with minor to no vertical displacement at the surface and shallow subsurface may indicate deep strike-slip faults (Fig. 2a).

### *Geological section along the LYG*

185 Borehole information from the Meizar and Mukheibeh wells, drilled along the LYG close to the area of the Hammat Gader springs (Figs. 1; 4a) provides a unique opportunity to explore the complex faulting pattern along major parts of the gorge. Based on the lithological interpretation of well logs (IHS, (Margane and Hobler, 1994)) that were drilled along the LYG, a geological cross-section was constructed and validated by the lithological description of all other well-sections in the area (Fig. 4b). At least the

190 uppermost part of the B2-A7 aquifer, the Campanian B2 horizon is disclosed in all wells of the study

area. Hence, borehole data about its elevation and thickness provide good coverage in the study area and are the key to the current work (Table 1).

The base of B2 displays large elevation changes over small distances, e.g. in borehole MU-6 located between MU-JRV1 and MU-8 (Fig. 4a), at distances of 1.2 km and 0.6 respectively. Although the B2

horizon appears at similar elevations in boreholes MU-JRV1 (-408 m msl.) and MU-8 (-403 m msl.), it was encountered in well MU-6 considerably deeper (-480 m msl.). Even more distinct, between boreholes MU-7A, MU-4, and MU-2, the elevation of base B2 is varying over a horizontal distance of 40 m to an extent of 130 m. These differences result from faulting (Fig. 3).

Considering places where the B2 formation was fully penetrated (Table 1) only, the average thickness of

200 the chert-bearing B2 layer is 189±14 m. MU-8 well contains an exceptionally thick Senonian sequence exceeding 290 m. Though the Senonian sequence in the region is well known for its thickness variations (Rosenthal et al., 2000a; Rosenthal et al., 2000b; Rosenthal, 1972), such variations over short distances are exceptional and require a different explanation. It is therefore suggested that the unit is either strongly tilted or thickness was "doubled" in the drilling due to crossing a thrust fault.

In the study area, the thickness of Turonian beds (A7) is fairly uniform and ranges between 300 to 350 m. However, in borehole MU-JRV1, a section of about 700 m consisting of two repeating sequences of marly limestone and dolomitic limestone, is regarded to be a Turonian unit "doubled" by thrust faulting (Fig. 3).

### *Tracking fault paths*

Following the seismic interpretation, it is suggested that faults detected on the surface or at shallow depths in the study area are likely to indicate deep-seated strike-slip faults. Based on that hypothesis, faults shown on geological cross-sections in the western Ajloun area (Sahawneh, 2011) were considered the southern extension of the Lower Yarmouk Fault (LYF). At the southernmost end, the fault is branching out from the DST and continuing to the NE. The fault crosses the LYG west of the Hammat

Gader – Meizar – Mukheibeh area (Fig. 1). North of the gorge, the fault turns further eastwards following the outcropping fault-lineaments mapped by Michelson (1979). Another possibility is a northward continuation of the fault joining with the Nov Fault Zone (NFZ) (Shulman et al., 2004).

Another SE-NW strike-slip fault was drawn according to previous interpretation and current reinterpretation of seismic data. Although parts of this fault were previously mapped as normal faults,

our revised interpretation suggests that it is most likely a strike-slip fault (Mevo-Hama Fault, MHF). In between these two newly suggested strike-slip faults (MHF and LYF), there is a thrust fault which clearly stands out in two seismic lines, GP-3661 and GP-3662. ,

### Summary and Conclusions

A comprehensive reinterpretation and compilation of available geological and geophysical data across

the LYG at a local scale are presented. Former seismic analysis of the deep structure in the Golan Heights (Meiler, 2011; Shulman et al., 2004; Shulman and May, 1989) used deep borehole data from Jordan and Syria to study regional scale structures. However, in these studies, all rock units between the Turonian

and Pliocene basalts were presented as one undivided unit and some of the faults at the southernmost DS-3545 line were neglected. The shallow seismic survey conducted at Mevo Hamma area (Bruner and Dekel, 1989) has been originally interpreted simultaneously with preliminary results of Shulman and May (1989) and with no access to data from northern Ajloun. Since then, the Mevo Hamma data has not been used by any other study done in the area.

As the Mukheibeh well field has been further developed in later years more data of Upper Cretaceous has been made available alongside hydrological observations. The rejuvenation of hydrological studies of the LYG started with conceptual models (Roded et al., 2013; Siebert et al., 2014), which speculated about potential reasons for local hydraulic anisotropies in order to explain the complex hydrological system manifested by groundwater pressure, temperature and chemical variations. This was followed by a series of numerical 2D models (Goretzki et al., 2016; Magri et al., 2015; Tzoufka et al., 2018), which used a combination of previous works to construct their structural model. The results achieved by those studies simulate groundwater flow and heat transfer over a fault in the geological sequence of the area. However, the transition of a 2D model into semi-3D (Magri et al., 2016) emphasized the problem of numerically simulating groundwater dynamic processes using a speculated structural model and the need for a unified structural solution in a complex setting like the LYG.

While tectonic models of pull-apart basins have been developed for different scales and various places around the world (e.g. Corti and Dooley, 2015; Sagy and Hamiel, 2017; Sugan et al., 2014) including models for the Dead Sea basin itself (Smit et al., 2008; 2010) they mostly address the shape of basin boundaries and internal processes. The unique situation at the eastern rim of the Kinnarot pull-apart basin includes the collision of the Golan Heights at the Hermon Mountain. To the best of our knowledge, such a scenario has not yet been modeled. Hence, the suggested fault system does not entirely comply with the findings of previous research on the surrounding tectonics of pull-apart basins performed in other tectonic settings. It is suggested that this mismatch may result from the settings at the eastern rim of the basin where the en-echelon changes from a chain of pull-apart basins to a push-up ridge.

Using available geological and geophysical data (Fig. 1), from the southern Golan and northern Ajloun, a new fault pattern has been delineated across the Lower Yarmouk Gorge. It includes a series composed of strike-slip and thrust faults, which may be associated with the regional Dead Sea Transform system and with the local Kinnarot pull-apart basin. It seems that these compressional and tensional structures have been developed to form a series of fault-blocks, causing a non-uniform spatial hydraulic connection between them.

The main motivation for the work presented here was to provide fundamental structural and geological information, which may have hydrological implications. It is common that structural features do affect groundwater movement. To develop management strategies for a reservoir, which is particularly important in areas of water scarcity and transboundary water resources, transient 3D flow simulations of the resource are the most appropriate solution to understand reservoir behavior. However, they must be based on realistic geometry, including structural features. Though formulated for the LYG, the study is intended to show the importance of such studies in providing the necessary database in geologically stressed areas without sufficient data.

**Acknowledgment**

This research was supported by the German Science Foundation DFG (grant MA4450/2) within the special program to support peaceful development in the Middle East. We greatly acknowledge the DFG
for their continuous support. We thank Eyal Shalev and Miki Meiler for sharing seismic data of southern Golan Heights. We thank Dr. Armin Margane and two anonymous reviewers for their very helpful comments, which significantly improved the quality of the paper.

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

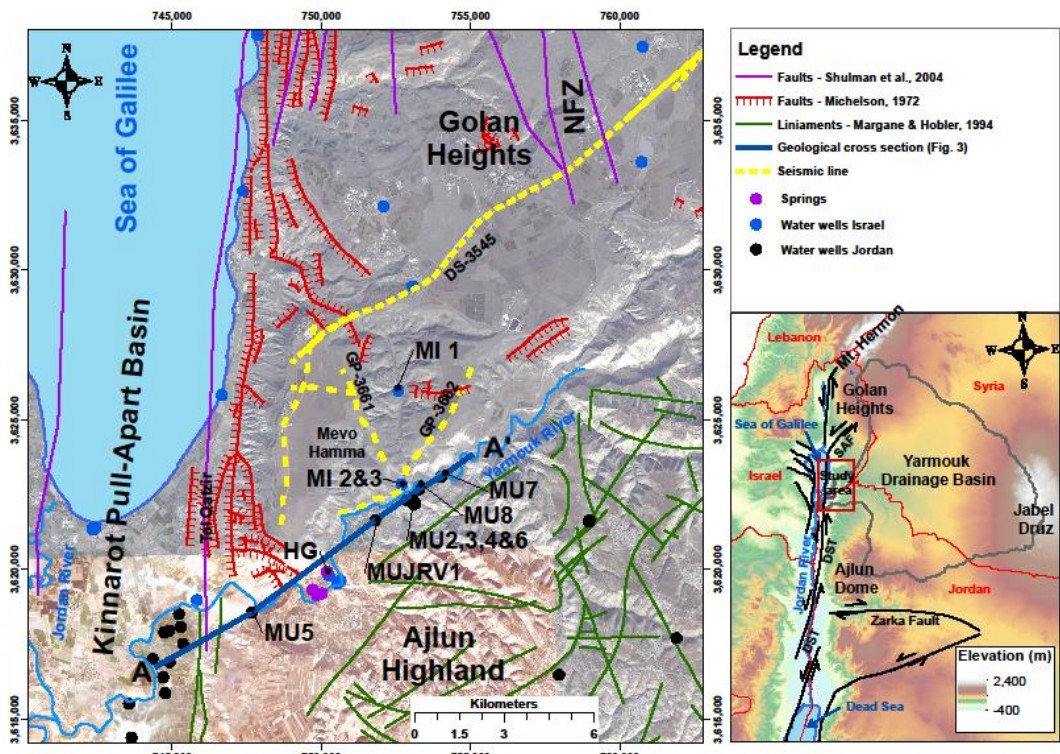

**Figure 1:** Regional location map on top of a digital elevation model, presenting the faults at the eastern rim of the Dead Sea Transform(DST), Sheikh Ali Fault (SAF), Nov Fault Zone (NFZ), wells belonging to Meizar (MI) and Mukheibeh (MU) and the location of the Hammat Gader (HG) springs. The location of seismic lines (DS-3543 and GP-3662; Fig. 2) and of a geotectonic cross-section along the LYG (A-A'; Fig. 3) is indicated as well.

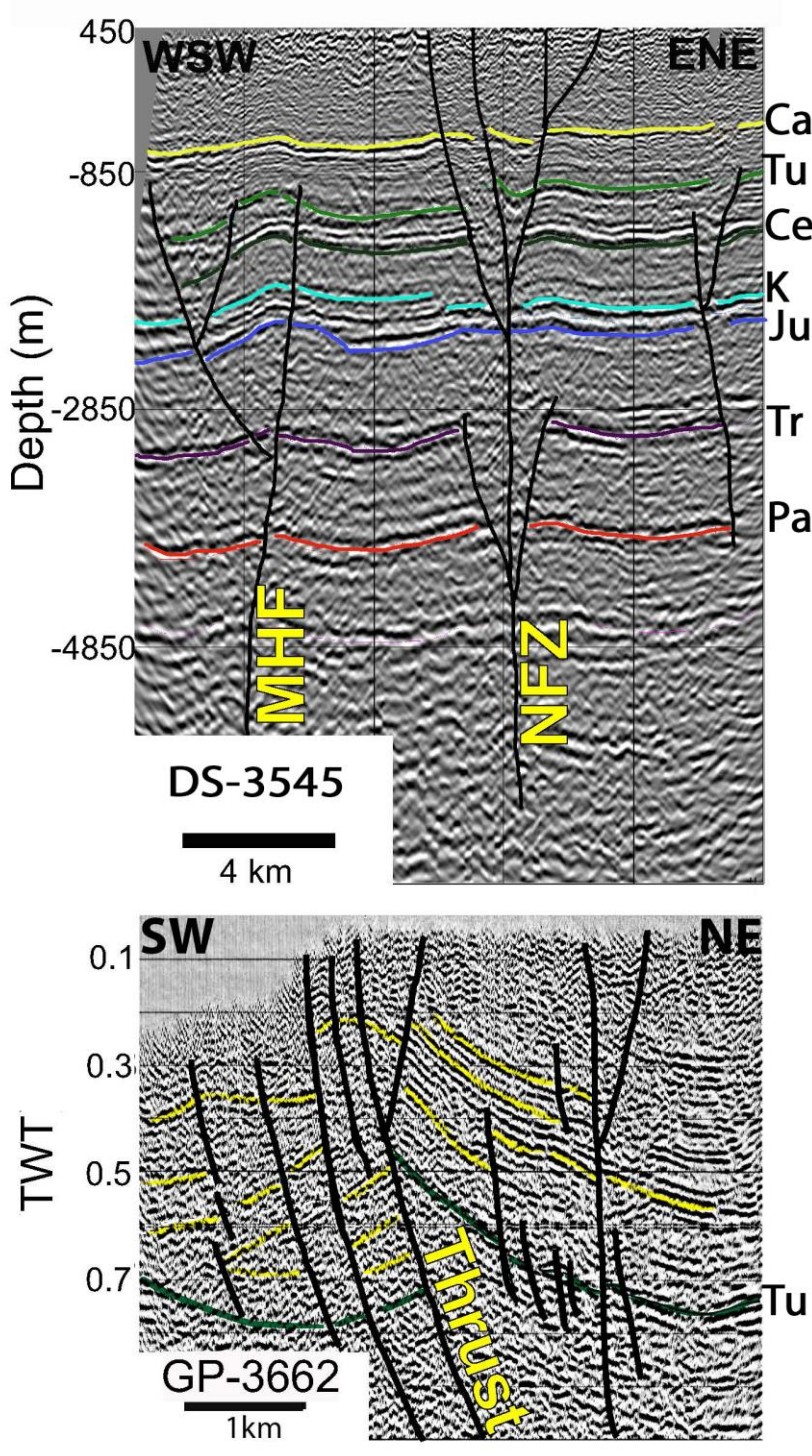

**Figure 2:** Seismic lines at Southern Golan. (a. top) seismic line DS-3545 showing flower-structure reinterpreted after Meiler (2011). Vertical scale in meters. (b. bottom) seismic line GP-3662 showing the thrust fault reinterpreted after (Bruner and Dekel (1989)). Vertical scale in two ways time.



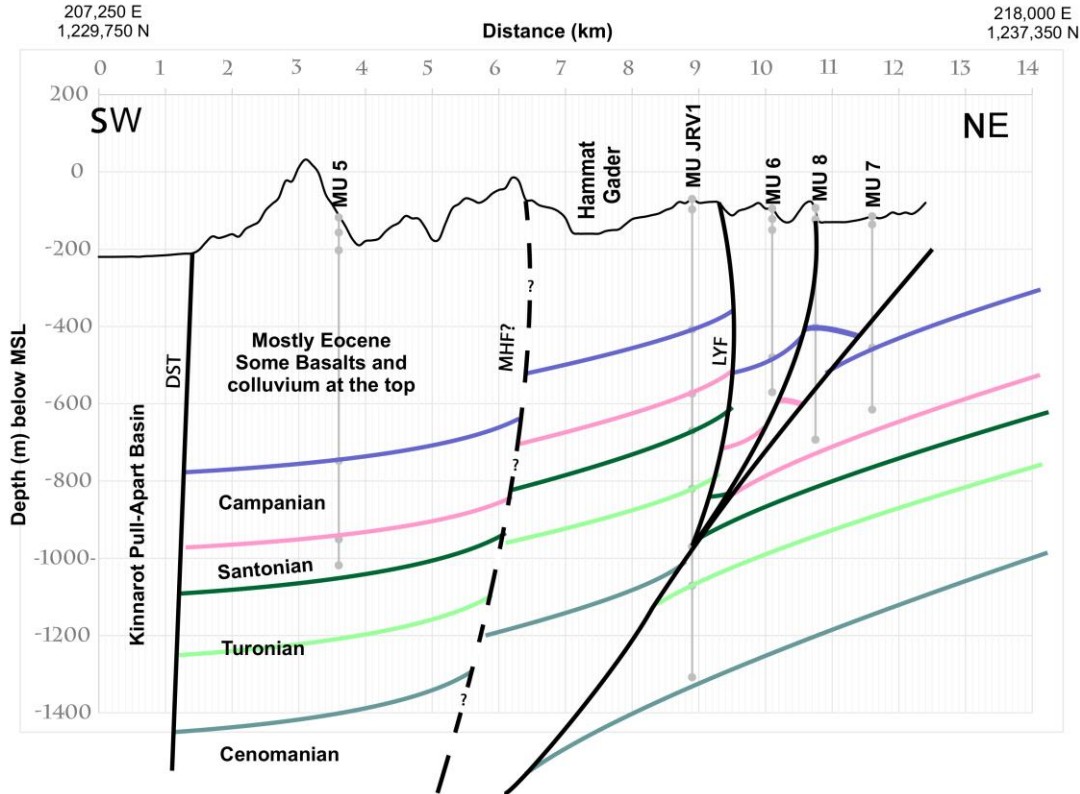

**Figure 3:** Geotectonic cross-section along the Lower Yarmouk Gorge, showing the interpretation of seismic line data and borehole information, which result in the given faults and depth position of geological formations.

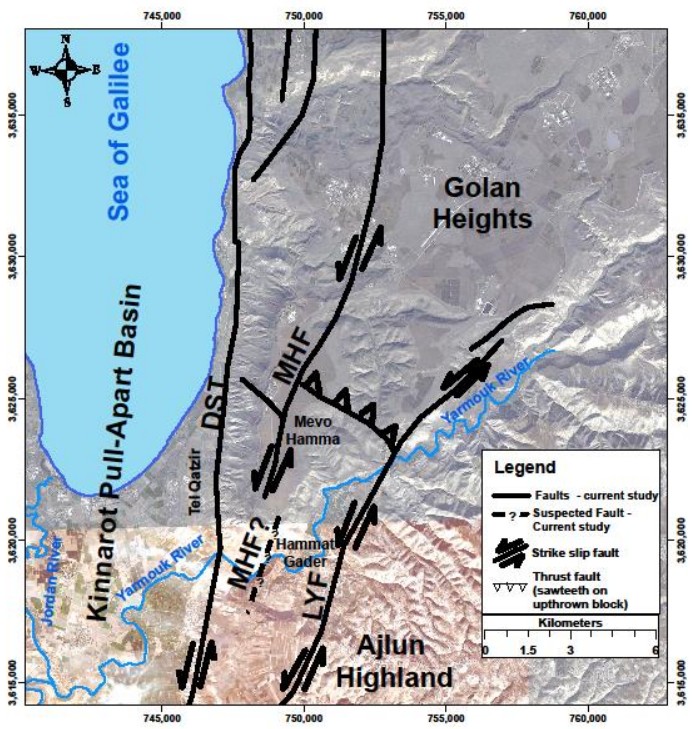

**Figure 4:** Results of the current study in a map local frame similar to the one used in Fig. 1. The proposed faults are at the area of the Lower Yarmouk Gorge, east of Kinnarot pull-apart basin, Mevo Hamma Fault (MHF) and Lower Yarmouk Fault (LYF). It is suggested that those faults results from the settings at the eastern rim of the DST at the Golan Heights where the en-echelon change from pull-apart basins to a push-up ridge at the Hermon Mt. (Fig. 1).

**Table 1:** Campanian chert bearing formation (B2) - thickness and elevation in wells adjacent to the LYG.

| Well name | Code | drilled lithology | Thickness (m) | Top (m MSL) | Underlying lithology |
|---|---|---|---|---|---|
| Mukheibeh 2* | MU2 | chert, limestone | 128 + | -470 | not penetrated |
| Mukheibeh 3* | MU3 | chert, limestone | 75 + | -338 | not penetrated |
| Mukheibeh 4* | MU4 | chert, limestone | 174 | -450 | limestone, dolomitic limestone |
| Mukheibeh 5* | MU5 | chert, limestone | 203 | -748 | limestone |
| Mukheibeh 6* | MU6 | chert, limestone | 90 + | -480 | not penetrated |
| Mukheibeh 7* | MU7 | chert, limestone | 160 + | -455 | not penetrated |
| Mukheibeh 8* | MU8 | chert, limestone | 290 + | -403 | not penetrated |
| Mukheibeh JRV1* | MU JRV1 | chert, limestone | 166 | -408 | limestone |
| Meizar 1 ** | MI1 | chert, chalk, limestone, marl | 195 | -649 | limestone, chalky limestone |
| Meizar 2 ** | MI2 | chert, chalk, limestone, marl | 210 | -424 | limestone, chalky limestone |
| Wadi Al Arab 4* | WA4 | limestone + chert | 182 | | limestone |
| Wadi Al Arab 1* | WA1 | limestone + chert | 193 | | limestone |
| Wadi Al Arab 2* | WA2 | limestone + chert | 180 | | limestone |
| Wadi Al Arab 5* | WA5 | limestone + chert | 148 + | | limestone |
| Douqara 1* | D1 | chert, marl, marly limestone, bitumineous shale at the top | 196 | | dolomitic limestone |

- *Source of information: Margane and Hobler (1994) available via DAISY, 2017;
- ** Source of information: well log

+ Partial thickness, not fully penetrated unit
