# Peer review of "Faulting patterns in the Lower Yarmouk Gorge - potentially influence groundwater flow paths"

_Hydrology and Earth System Sciences, 2018_

## Referee Comment (RC1) · Anonymous Referee #1 · 24 May 2018

Major comment: On figure 4, it seems there are no faults leading to the springs of Hamat Gader. Please explain better how water reach those springs? or suggest an alternative strategy to overcome this issue. Minor comments: Line 20: suggest change 'were previously' to 'were so far' or 'remain until now' Line 22: reads like the fault was 'constructed by' you… should be 'followed a compilations and revisions of…' p.3, and Line 110: the Authors have clearly tried to avoid the notion of Israel and Jordan, yet, as observed in the top paragraph this task is not viable. Please add the Israel and Jordan at least to the introduction and map 1.

L126: Fig 3 shows map not cross section.

References: - Suggest to add: Starinsky, A., Kats, A., Levite, D. (1979). Temperature-composition-depth relation in rift Valley hot springs Hamat Gadder, northern Israel.

[Figure]

Chem. Geol. 27, 233-244

Table 1: - add row for short name as appeared in Figures. - Standard the capitalization on lithology row. Fig 2: - Perhaps add the in-interpreted section as well in the time domain? - On the interpreted sections add units name or numbers (e.g. for Turonian, Jurassic etc) Fig 3 and 4a: - Seems overlapping. Maybe choose on them? Fig. 4: - Split to two figures- one for the map other for the cross section, and give proper titles. - change W-E to SW-NE - I found it strange that the markers are faulted (shifted vertically) , yet you state there is no fault.

[Figure]

---

## Referee Comment (RC2) · A. Margane (Referee) · 30 May 2018

nice compilation of facts, except those on the Syrian side where it should be mentioned that there are no known seismic profiles available (for this it could be referred to BREW et al., 2010 (Tectonic and Geologic Evolution of Syria, GeoArabia, Vol. 6, No. 4); which has e.g. Bussra-1 borehole, also mentioned in Meiler's PhD). It should also be added that on the Jordanian side there are no seismic profiles in this area. The closest oilwell is NH-1. By the way, you might also want to refer to geol structure contour maps for Northern Jordan done by me in the mid 1990s (Margane & Hobler, 1994) and available by your team for the SMART project. In the abstract and later in the text it should be made clear that the new fault system was not inferred from remote sensing. In fact that's the disturbing part: "your" faults are not even located in the valleys/topo lows. So

[Figure]

I am wondering how could they be inferred in areas where there is no geophysics like in NW Jordan ? Knowing how imprecise the fault system of Jordan was mapped by NRA in particular in the old days when satellite images where not used (until the 1990s) and when there was a lot of shifting in geological maps also due to the way mapping was conducted (we are speaking about hundreds of meters), I wouldn't dare using these as reference. 119-21: there is no proof for these faults. 123: relies Chapter 4: explain details of seismic data acquisition, e.g. spacing Text/graphics (Figures 2/3/4): text uses DS-3545, graphics use DS-3543; plz correct whichever is wrong Chapter 5: Would be good to add mentioned locations, like Hamat Gader spring and Meizar 2 well in Fig. 3 219: why should faults be a constraint for GW modelling ? Figure 1: you might want to add SF-Siwaqa Fault Figure 2: plz add in Figure 3 which part of that seismic profile is shown here Table 1: I have doubts Daisy is a valid reference. Can the described data be accessed by anyone (not being eligible) ? Figure 3/4: add that the cute green lines are adopted from NRA geological maps

MARGANE, A. & HOBLER, M. (1994): Groundwater Resources of Northern Jordan, Vol.3: Structural Features of the Main Hydrogeological Units in Northern Jordan. - Technical Cooperation Project 'Advisory Services to the Water Authority of Jordan', BGR & WAJ, BGR archive no. 118702:1-3, 57 p., 30 app., 38 ann.; Amman.

---

## Referee Comment (RC3) · Anonymous Referee #3 · 6 Jun 2018

"Faulting patterns determining groundwater flow paths in the Lower Yarmouk Gorge" by Inbar et al., presents a compilation of all available geological and geophysical data from the lower Yarmouk Gorge area. These include borehole data, geology and seismic cross-sections from south of the Sea of Galilee to the southern Golan Heights and including some data from Jordan. Results present a new faulting pattern for the area east of the Sea of Galilee. The authors suggest the presence of strike-slip faults, which form fault blocks and control hydrological parameters in the region.

While the paper presents an important reinterpretation and compilation of data within a regional context and adds information on the fault pattern along the northern section of the southern Dead Sea fault, there are some problems. The main issue I have is the lack of a proper, comprehensive discussion. Currently, the discussion is mixed in with

the results of the study. By separating them into two individual sub-chapters, both will benefit. The Results section needs to be expanded and presented in a more precise and focused way.

The discussion must go into more critical issues that are currently lacking such as: - Tectonic implications - Hydrological implications. In a paper titled "Faulting patterns determining groundwater flow paths in the Lower Yarmouk Gorge" – you do not discuss groundwater flow patterns - A discussion of the mechanism that formed these faults - do they fit in with what we know about the stress field in the area? It is not enough to say that since they are along the DSF, it fits. . . - Also, satatemets such as: "The present study shows that although there is no evidence for large vertical displacements, strike-slip faults must cross the LYG forming fault-blocks. Therefore, these faults must be taken into consideration when discussing groundwater hydrology" (lines 118-120) need to be formulated in a more rigorous way. Why "must" strike slip faults cross the LYG?

In addition, the discussion should address issues brought up in the Introduction. On lines 56-64 you mention the hypothetical fault trace of Magri et al. (2016), yet you do not refer back to this to show if your results support this trace or not. This is also true for the results of Goretzke et al (2016), which you mention in the introduction – do your results support or disprove their theories?

The GII provides a comprehensive database of historical-recent seismicity. In such a study, I would also expect you to use seismicity to prove the presence of strike-slip faults or thrusts and help rule out previous suggestions. I think this would strengthen your arguemnts.

Technical issues: - While the paper is well written, there are still a few grammatical mistakes. Please check the English again. - Every location, borehole, seismic line, etc. mentioned in the text needs to appear on a comprehensive location map, which is referred to in the text (e.g. Meizar 1,2 & 3 (Fig. 1)). This should be the very first figure of the manuscript (and not the third). This holds for every place name mentioned

in the introduction and throughout the paper (GH, LYG, Golan syncline, Mt. Hermon, Sheikh Ali fault, Ajloun Dome, Hammat Gader and Mukheibeh springs, Meizar 1,2 & 3...) - Line 19-20: Please do not cite in the Abstract - Line 31: The more correct term is "Dead Sea fault (DSF)" in keeping with the "San Andreas fault" - Line 97-98: The DST was already defined above. There is no need to define it again. - Lines 99-100: You present the eastern fault entering the Seaof galilee as the main branch of the DSF. Why do you rule out the western branch (Hurwitz et al., 2002)? - Line 134: "dots were connected" – please use a more scientific term. Perhaps "interpolation was carried out between data points" - Lines 140-146: From this paragraph it seems that you use the interpretation of Meiler (2011) for seismic cross section DS-3545. So when you say "reinterpretation" on line 149, what do you mean? It is not clear if you just took Meiler's interpretation or if you did something of your own. Please clarify. - Line 151-152: I do not understand the logic of the argument. Please rephrase so that it is clearer. Why is a thrust fault the more logical solution? - Line 156: Seismic data is not measured. It is collected. - Line 159: Repetition - Line 184-187: Please refer to Figure 4 - Lines 191-193: Repetition

Figures: Figure 1 should be a comprehensive location map that includes all places, boreholes, seismic lines, features, etc. mentioned in the text. Figure 2: please show uninterpreted seismic line together with the interpretation. Also on 2b – the vertical scale cannot be depth since this is a time section. Figure 3: this is in fact your location map and should come first. I cannot see the difference between Figure 4a and the bottom of Figure 3. Why do you need both if they are the same except for the location of profile A-A'?

---

## Author Comment (AC1) · 30 Jul 2018

**Reviewer #1**

Dear reviewer 1,

We thank you for your valuable comments, which very much helped to improve clarity of the manuscript! We hope to reply in the following sufficiently to your remarks.

**C1: Major comment: On figure 4, it seems there are no faults leading to the springs of Hamat Gader. Please explain better how water reach those springs? or suggest an alternative strategy to overcome this issue.**

A1: This is by all mean an important question. The aim of the current MS is to delineate the main fault block system which controls the different hydrological observations in the area. However, those blocks exhibit extensive inner faults as was clearly described by Bruner and Dekel, 1989. This complexity can be seen also on the geological profile (Fig. 3) where only the main stem of the LYF is shown on the map (Fig. 1). Due to the limited available data, we receive only a coarse picture of the subsurface and hence, our interpretation "provides a **coarse** fault block model..." as mentioned in the abstract - line 30

**C2: Minor comments: Line 20: suggest change 'were previously' to 'were so far' or 'remain until now'**

A2: We follow the suggestion and changed it.

**C3: Line 22: reads like the fault was 'constructed by' you. . . should be 'followed a compilations and revisions of. . .'**

A3: We follow the suggestion and changed it.

**C4: p.3, and Line 110: the Authors have clearly tried to avoid the notion of Israel and Jordan, yet, as observed in the top paragraph this task is not viable. Please add the Israel and Jordan at least to the introduction and map 1.**

A4: We included respective information. In map 1, the international borders are included and respective country names are given.

**C5: L126: Fig 3 shows map not cross section.**

A5: After amending the figures we corrected all figures references.

**C6: References: - Suggest to add: Starinsky, A., Kats, A., Levite, D. (1979). Temperature composition-**

depth relation in rift Valley hot springs Hamat Gadder, northern Israel. Chem. Geol. 27, 233-244

A6: We follow the suggestion and changed it.

**C7: Table 1: - add row for short name as appeared in Figures. - Standard the capitalization on lithology row.**

A7: We follow the suggestion and changed it.

**C8: Fig 2: - Perhaps add the in-interpreted section as well in the time domain? - On the interpreted sections add units name or numbers (e.g. for Turonian, Jurassic etc)**

A8: Adding the un-interpreted GP-3662 section is not possible as the report by Brunner and Dekel (1989) contain only stack (un-interpreted) and migrated (interpreted) sections. The presented section is (as described in the caption)

reinterpretation after Bruner and Dekel. Unit names were omitted by mistake in the current version – We follow the suggestion and changed it.

**C9: Fig 3 and 4a: - Seems overlapping. Maybe choose on them? Fig. 4: - Split to two figures- one for the map other for the cross section, and give proper titles. - change W-E to SW-NE.**

A9:  We follow the suggestion and changed it.

**C10: I found it strange that the markers are faulted (shifted vertically), yet you state there is no fault.**

A10: Based on the available interpreted data in the study area, we are not able to trace a fault along the path of the gorge. Unpublished data by GSI indicate a SW-NE running fault, coming from the Lower Jordan Valley and reaching Hammat Gader indicate, faulting may also have occurred along the gorge. However, according to our interpretation, several faults crosses the gorge but - as far as observable - faults exhibits little to no vertical displacement at the surface.

---

## Author Comment (AC2) · 30 Jul 2018

Reviewer #2

Dear Dr. Margane,

Particularly due to your long-term expertise in the area, we highly appreciated your kind words and constructive remarks on the manuscript! We hope to meet your points in the following.

**C1:  on the Syrian side where it should be mentioned that there are no known seismic profiles available (for this it could be referred to BREW et al., 2010 (Tectonic and Geologic Evolution of Syria, GeoArabia, Vol. 6, No. 4); which has e.g. Bussra-1 borehole, also mentioned in Meiler's PhD). It should also be added that on the Jordanian side there are no seismic profiles in this area. The closest oilwell is NH-1.**

A1: The aim of the presented study is to delineate the structural guidelines across the Yarmouk Gorge at the area of Hammat Gader – Meizar – Mukheibeh. Therefore, Syrian data are not discussed. Similarly, oil wells and remote seismic lines in Jorden and Israel are not discussed.

**C2: By the way, you might also want to refer to geol. structure contour maps for Northern Jordan done by me in the mid 1990s (Margane & Hobler, 1994) and available by your team for the SMART project.**

A2: Of course, we only missed that citation. Corrected.

**C3: In the abstract and later in the text it should be made clear that the new fault system was not inferred from remote sensing.**

A3: In that paper we have not dealt with methods such as remote sensing, gravity, magnetics, etc., and therefore those methods are not discussed.

**C4: In fact that's the disturbing part: "your" faults are not even located in the valleys/topo lows. So I am wondering how could they be inferred in areas where there is no geophysics like in NW Jordan?**

A4: The faults described in the manuscript were inferred from seismic lines at the Golan Heights and their southward continuation across the gorge was inferred by geological profiles. One of the profiles was constructed as part of the current study, others were published by Sahawneh (2011). A line indicating the northernmost profile by Sahawneh was added to the map.

**C5: Knowing how imprecise the fault system of Jordan was mapped by NRA in particular in the old days when satellite images where not used (until the 1990s) and when there was a lot of shifting in geological maps also due to the way mapping was conducted (we are speaking about hundreds of meters), I wouldn't dare using these as reference.**

A5: We are aware of the critical correctness of the old geological maps, available from NRA. The original maps have been made available as GIS files, which have been corrected by Julia Sahawneh and own work at the UFZ.

**C6: 119-21: there is no proof for these faults.**

A6: That's correct, we reshaped the sentence.

**C7:** **123: relies Chapter 4: explain details of seismic data acquisition, e.g. spacing Text/graphics (Figures 2/3/4): text uses DS-3545, graphics use DS-3543; plz correct whichever is wrong.**

**A7:** general description of seismic data collection was added with references to works with detailed description of the various parameters. DS-3545 is correct, typo was corrected.

**C8:** **Chapter 5: Would be good to add mentioned locations, like Hamat Gader spring and Meizar 2 well in Fig. 3**

**A8:** According to the remarks of RC1 we amended figures and new fig. 1 includes all information

**C9:** **219: why should faults be a constraint for GW modelling?**

**A9:** Faults are often regarded as relative flow barriers with different hydraulic properties from the surrounding rock. Depending on the specific properties of the fault, it may either block or divert GW flow. In both cases it is constraining the model.

**C10:** **Figure 1: you might want to add SF-Siwaqa Fault . Figure 2: plz add in Figure 3 which part of that seismic profile is shown here**

**A10:** all maps were joined into a new Figure (Fig.1), which does not show any more Siwaqa Fault. The figure show the entire length of the seismic lines presented in Fig. 2

**C11:** **Table 1: I have doubts Daisy is a valid reference. Can the described data be accessed by anyone (not being eligible)?**

**A11:** We included the original reference Margane and Hobler, 1994, which is made available via DAISY. Daisy is accessible at http://www.ufz.de/daisy and holds data from several sources.

**C12:** **Figure 3/4: add that the cute green lines are adopted from NRA geological maps MARGANE, A. & HOBLER, M. (1994): Groundwater Resources of Northern Jordan, Vol.3: Structural Features of the Main Hydrogeological Units in Northern Jordan. - Technical Cooperation Project 'Advisory Services to the Water Authority of Jordan', BGR & WAJ, BGR archive no. 118702:1-3, 57 p., 30 app., 38 ann.; Amman.**

**A12:** Yes, please see answer on that comment above. Corrected also in the new Fig. 1.

---

## Author Comment (AC3) · 30 Jul 2018

Reviewer #3

Dear Reviewer 3,

We thank you a lot for these critical remarks, by which the quality of the paper was much improved! We hope to sufficiently reply to your comments in the following.

**C1:** **The main issue I have is the lack of a proper, comprehensive discussion. Currently, the discussion is mixed in with the results of the study. By separating them into two individual sub-chapters, both will benefit. The Results section needs to be expanded and presented in a more precise and focused way. The discussion must go into more critical issues that are currently lacking such as: - Tectonic implications - Hydrological implications.**

A1: This comment, which refers to the structure of the manuscript was seriously considered. We have tried to transform Chapter 5 (Results) into results and discussion chapters as suggested. Those attempts, has further strengthen our initial consideration that the discussions parts of the chapter are in most cases supporting the rational presentation of results. However, the point raised is absolutely clear and therefore we have transformed the "Conclusion" chapter (chapter 6) into "Summary and conclusion". We hope that this new chapter answers most of the points raised in regards to the order and clarity of the presentation. With that said and done another point has to be clarified, it is not the scope of the current research to provide all answers regarding implications. The current manuscript suggests a new structural framework for future hydrological research together with questions that will hopefully lead to additional tectonic study and subsequently a much more advanced understanding of the geotectonic in that complex area.

**C2:** **In a paper titled "Faulting patterns determining groundwater flow paths in the Lower Yarmouk Gorge" – you do not discuss groundwater flow patterns - A discussion of the mechanism that formed these faults - do they fit in with what we know about the stress field in the area?**

A2: The aim of the presented research is to provide additional information to the structural framework of the area, which doubtless implies the geohydrological systems, as suggested by Tzoufka et al., 2018. It is correct, groundwater flow pattern is neither investigated nor discussed here.

The structure is described following interpretation of geological and geophysical data/findings and not following a study of kinematics, which may resolve once the questions of faulting mechanism.

That point is substantially important and we hence changed the title of the manuscript accordingly.

**C3:** **It is not enough to say that since they are along the DSF, it fits. Also, statements such as: "The present study shows that although there is no evidence for large vertical displacements, strikeslip faults must cross the LYG forming fault-blocks. Therefore, these faults must be taken into consideration when discussing groundwater hydrology" (lines 118-120) need to be formulated in a more rigorous way. Why "must" strike slip faults cross the LYG?**

A3: Thank you for that remark, we changed it to "might be related" and "suggested to consider".

**C4:** **In addition, the discussion should address issues brought up in the Introduction. On lines 56-64**

**you mention the hypothetical fault trace of Magri et al. (2016), yet you do not refer back to this to show if your results support this trace or not. This is also true for the results of Goretzke et al (2016), which you mention in the introduction – do your results support or disprove their theories?**

A4: This remark contributed to the new chapter "summary and conclusion", thank you! We agree with the work of Magri et al. (2015); Goretzki et al. (2016) and Tzoufka et al. (2018) that simulate groundwater flow and heat transfer across a fault crossing the lithological sequence. However, due to the limited available data, we receive only a coarse picture of the subsurface. Based on these available data in the study area, we are not able to trace a fault along the path of the gorge. However, unpublished data by GSI indicate a SW-NE running fault, coming from the Lower Jordan Valley and reaching Hammat Gader indicate, faulting may also occurred along the gorge, which would directly support the cited hydrological modeling studies

**C5: The GII provides a comprehensive database of historical-recent seismicity. In such a study, I would also expect you to use seismicity to prove the presence of strike-slip faults or thrusts and help rule out previous suggestions. I think this would strengthen your arguments.**

A5: We have scanned that database for all its content (about 100 years). We have not found any event with epicenter located in the study area. We can only conclude that the system has not been active for the last 100 years. As we believe that such statement does not contribute to the presentation it was omitted from the final text.

**C6: Technical issues: - While the paper is well written, there are still a few grammatical mistakes. Please check the English again.**

A6: We followed the advice and corrected grammatical mistakes.

**C7: Every location, borehole, seismic line, etc. mentioned in the text needs to appear on a comprehensive location map, which is referred to in the text (e.g. Meizar 1,2 & 3 (Fig. 1)). This should be the very first figure of the manuscript (and not the third). This holds for every place name mentioned in the introduction and throughout the paper (GH, LYG, Golan syncline, Mt. Hermon, Sheikh Ali fault, Ajloun Dome, Hammat Gader and Mukheibeh springs, Meizar 1,2 & 3)**

A7: Following that remark we have changed figure 1 to include also figure 3 and 4a. The new figure 1 has all required information. Thank you very much this step dramatically improved the MS.

**C8: Line 19-20: Please do not cite in the Abstract**

A8: Agree. Citation was removed from the abstract

**C9: Line 31: The more correct term is "Dead Sea fault (DSF)" in keeping with the "San Andreas fault"**

A9: The term DST is well known and accepted in the literature. Google scholar gives about 359,000 results for that term, starting with Garfunkel, Zvi. "Internal structure of the Dead Sea leaky transform (rift) in relation to plate kinematics." *Tectonophysics*80.1-4 (1981): 81-108. We prefer to keep this term.

**C10: Line 97-98: The DST was already defined above. There is no need to define it again.**

A10: Agree. Changed

**C11: Lines 99-100: You present the eastern fault entering the Sea of Galilee as the main branch of the DSF. Why do you rule out the western branch (Hurwitz et al., 2002)?**

A11: we accept this suggestion and added faults on the western side of the lake to complete the picture. Still, it is important to note that the N-S fault on the western side is not a strike-slip fault but an oblique-slip fault (Inbar, 2012)

**C12: Line 134: "dots were connected" – please use a more scientific term. Perhaps "interpolation was carried out between data points"**

A12: Thank you, we change it to more scientific terminology.

**C13: Lines 140-146: From this paragraph it seems that you use the interpretation of Meiler (2011) for seismic cross section DS-3545. So when you say "reinterpretation" on line 149, what do you mean? It is not clear if you just took Meiler's interpretation or if you did something of your own. Please clarify.**

A13: It is repeatedly mentioned that we used Meiler's PSDM processes line and added our interpretation. The new interpretation adds new faults, which were previously neglected by Shulman (1989 and 2004) as well as by Meiler (2011). Those newly interpreted faults were probably not important for the regional study however they are highly significant when studying the LYG. We have also amended horizon identification, however acceding the scope of the current MS this topic is not discussed.

**C14: Line 151-152: I do not understand the logic of the argument. Please rephrase so that it is clearer. Why is a thrust fault the more logical solution?**

A14: Rephrased. "Another possible solution for the 2D seismic data is a thrust fault. This solution seem to fit better with the newly presented structural frame."

**C15: Line 156: Seismic data is not measured. It is collected.**

A15: Thank you, it's corrected.

**C16: Line 159: Repetition**

A16: Thanks. Deleted

**C17: Line 184-187: Please refer to Figure 4**

A17: Thanks, done.

**C18: Lines 191-193: Repetition**

A18: Yes, the entire paragraph is rephrased.

**C19: Figures: Figure 1 should be a comprehensive location map that includes all places, boreholes, seismic lines, features, etc. mentioned in the text.**

A19: Yes, it has been changed accordingly.

**C20: Figure 2: please show non-interpreted seismic line together with the interpretation. Also on 2b – the vertical scale cannot be depth since this is a time section.**

A20: yes, that's correct – we changed it.

**C21: Figure 3: this is in fact your location map and should come first. I cannot see the difference between Figure 4a and the bottom of Figure 3. Why do you need both if they are the same except for the location of profile A-A'?**

A21: We changed it entirely.

---

## Referee Report (RR1)

[referee-annotated manuscript omitted]

---

## Referee Report (RR2)

C1: Please fix the English.

A1: First, we would like to thank the anonymous referee for improving the manuscript and correcting many of our English mistakes. Corrected.

C2: During the last 10 years there has been a strong movement in the geological community to change this to confirm with worldwide nomenclature – it is the San Andreas fault, the east Anatolian fault, the Alpine fault and therefore, the Dead Sea fault. **I will not insist**, but it is a point you should be aware of.

A2: Thank you for that remark. We are fully aware of that movement and the importance of proper use of the terminology. We have considered "Dead Sea fault (DSF)" or "Dead Sea fault zone (DSFZ)" however, the "Dead Sea Transform (DST)" is a term that seems to us more appropriate as the tectonic discussion deals with the pull-apart basin and the left lateral movement on the transform.

C3: There are two small issues that need to be addressed (lines 157-158; 180). Please see my comments on the annotated manuscript.

A3: We follow the suggestion and changed it.

C4: I am missing a final figure that summarizes the findings.

A4: All findings and end results found in figure 1.

C5: The main concern I have is that I still find the article lacking in strong implications. So it is a new fault map and I agree that it has hydrological consequences, but at the end of the day, it is still just a local fault map. If you want to publish in an international journal, it should have some more far-reaching implications. I understand the scope of the research, but why would the article interest someone in the United States? Why should someone in Australia read it? It is missing something fundamental to take it to that next level. It is really not that difficult to do – a few sentences in the introduction and a few more in the conclusions – If areas where strike-slip meet collision haven't been modeled or studied (lines 231-235), use this to your advantage...

A5: Thank you for that important remark. We followed the suggestion and elaborated on the unique tectonic settings at the eastern rim of the Kinnarot pull-apart basin.

However, we feel it is important to note here that this is not "just a local fault map", it is a suggested solution for an enigmatic hydrological situation concerning three riparian states in a water scarce region. In areas of water scarcity and transboundary water resources, transient 3D flow simulations of the resource are the
 most appropriate solution to understand reservoir behavior. This is an important tool for the development of management strategies. However, those models must be based on realistic geometry, including structural features. And although we work at the LYG, the study is intended to show the importance of such kind of structural investigations to provide the necessary database in geologically stressed areas without sufficient data. Furthermore, during the hydrogeological investigation, we have discovered a mismatch to rim faults evolution as studied by common pull-apart basin models. We argue that this mismatch may result from the settings at the eastern rim of the basin as the en-echelon change from pull-apart basin to a push-up ridge. We thank the referee for insisting on emphasizing that important note.

[revised manuscript text omitted]

---

## Author Response (AR3)

5 Anonymous Referee #3

C1: The way the authors formulated the importance of this study in the letter addressing my comments is exactly what I was looking for. This appears in the final text, but in with weaker wording. Please consider adding your response to me instead. It is well written and clearly stresses the point you are trying to make.

10 A1: Thank you for that comment. Although it appears in places throughout the text, your remark emphasized that it should be better highlighted. Therefore, slightly changed, that text was added to the abstract.

C2: Figure 1 - needs to be separated into two figures - a location map showing all the names of places, boreholes, seismic lines etc mentioned in the text and a final figure showing the results of this study. They cannot be on the same figure. the first should be in the beginning
15 and the second near the end so we can be convinced by the rest of the figures and the text.

A2: Accepted remark – we have changed the figure accordingly

C3: There is a discrepancy between the manuscript as it appears at the end of the "comments" letter and the "clean" manuscript uploaded to the system. This is evident in the new section near the bottom of page 3. Please make sure you submit the updated version.

20 A3: Should not have happened, will be double-checked this time.

C4: Also - the English of the new sections should be checked. I did my best to fix it, but I may have missed some things. Please see annotated manuscript for minor comments

A4: Thank you very much for your effort. Your work has clearly improved the shape of the MS and should have been better acknowledged.

[revised manuscript text omitted]